# Regulating the Expression of HIF-1α or lncRNA: Potential Directions for Cancer Therapy

**DOI:** 10.3390/cells11182811

**Published:** 2022-09-08

**Authors:** Minghui Zhang, Yu Zhang, Yubo Ding, Jialu Huang, Jingwei Yao, Zhuoyi Xie, Yufan Lv, Jianhong Zuo

**Affiliations:** 1The Laboratory of Translational Medicine, Hengyang Medical School, University of South China, Hengyang 421001, China; 2The Affiliated Nanhua Hospital of University of South China, Hengyang Medical School, University of South China, Hengyang 421002, China; 3The Third Affliated Hospital of University of South China, Hengyang Medical School, University of South China, Hengyang 421900, China

**Keywords:** hypoxia, HIF-1α, lncRNAs, EMT, HIF-1α inhibitors

## Abstract

Previous studies have shown that tumors under a hypoxic environment can induce an important hypoxia-responsive element, hypoxia-induced factor-1α (HIF-1α), which can increase tumor migration, invasion, and metastatic ability by promoting epithelial-to-mesenchymal transition (EMT) in tumor cells. Currently, with the deeper knowledge of long noncoding RNAs (lncRNAs), more and more functions of lncRNAs have been discovered. HIF-1α can regulate hypoxia-responsive lncRNAs under hypoxic conditions, and changes in the expression level of lncRNAs can regulate the production of EMT transcription factors and signaling pathway transduction, thus promoting EMT progress. In conclusion, this review summarizes the regulation of the EMT process by HIF-1α and lncRNAs and discusses their relationship with tumorigenesis. Since HIF-1α plays an important role in tumor progression, we also summarize the current drugs that inhibit tumor progression by modulating HIF-1α.

## 1. Introduction

Numerous studies have demonstrated that many solid tumors exist in a hypoxic microenvironment as a result of their fast proliferation and relatively insufficient blood supply. Tumor cells adapt to hypoxia on this basis by controlling angiogenesis, modifying cellular metabolic pathways, and altering pre- and post-transcriptional gene expression levels [1,2]. Notably, the production and accumulation of an important factor, HIF-1α, induced by tumor cells in hypoxia plays an important role in a range of regulatory responses in tumors [3]. Epithelial–mesenchymal transition (EMT) is an important factor in the development of tumor invasion, migration, and distant metastasis [4,5]. Overexpression of HIF-1α can promote the production and signal transduction of EMT transcription factors, thus promoting the EMT process of tumor cells and accelerating the invasion, migration, and metastatic ability of tumors [6,7]. Currently, with the increased research on non-coding RNAs, functional studies of lncRNAs are receiving increasing attention [8]. Many lncRNAs in cancer cells have been found to be regulated by HIF-1α, leading to altered expression levels, and changes in the levels of some of these lncRNAs can promote the progression of EMT in tumor cells [9,10]. Therefore, we need to further explore the mechanism by which overexpressed HIF-1α under hypoxic conditions regulates changes in lncRNA levels in tumor cells and how HIF-1α plays a role in promoting EMT, leading to tumor development and poor prognosis, and ultimately finding targets for anticancer therapy.

## 2. Induction of HIF-1α Accumulation in Tumor Cells under Hypoxia

Hypoxia-inducible factor (HIF) comprises two subunits: HIF-α and HIF-β, with HIF-α being able to be split into HIF-1α, HIF-2α, and HIF-3α [11]. Due to their distinct proline hydroxylase sites (Pro564 and Pro402 in HIF-1α, Pro405+ and Pro531 in HIF-2α), HIF-1α and HIF-2α may be susceptible to various levels of hypoxia [10]. HIF-1α accumulates at extremely low oxygen levels (0–2%) to assist cells in resisting transient hypoxia, whereas HIF-2α shows more persistent expression under moderate hypoxia (2–5%), and HIF-3α function is largely dependent on the control of other HIF complexes [12]. Of these, HIF-1 is the most widely studied heterodimeric complex composed of the HIF-1α subunit and HIF-1β subunit, existing in almost all cells [13]. Under hypoxic conditions, HIF-1 is involved in the transcriptional regulation of glycolytic genes such as glucose transporter proteins and glycolytic enzymes; HIF-1 also maintains the redox dynamic balance in tumor cells under hypoxic conditions by inhibiting fatty acid oxidation; in addition, HIF-1 can affect mitochondrial metabolism by regulating the production of reactive oxygen species and signal production in mitochondria [14]. Notably, the HIF-1α subunit and HIF-1β subunit must undergo binding to form a heterodimer before they can exert their biological functions [10]. HIF-1α contains multiple functional structural domains, including the basic helix−loop−helix (bHLH) structural domain, the Per-ARNT-Sim (PAS) structural domain, the oxygen-dependent degradation structural domain (ODD), and the trans-activation structural domain (TAD)s (Appendix A, Figure A1). The bHLH and PAS domains are involved in dimerization and DNA binding, while the ODD domain is mainly involved in the proteasomal degradation of HIF-1α under normoxic conditions, and the TAD domain is involved in the regulation of target genes by HIF-1α (Figure A1) [15]. Under normoxic conditions, translated HIF-1α is recognized by Von Hippel–Lindau (PVHL) proteins after hydroxylation of its proline residues (Pro-402 and Pro-564) by prolyl hydroxylase domain-containing enzymes (PHD), after which the PVHL protein recruits ubiquitin ligase to bind to HIF-1α, and the HIF-1α bound to ubiquitin ligase enters the proteasome to be crushed and degraded, eventually losing its biological function (Appendix A, Figure A2) [16]. Conversely, when the hydroxylation of HIF-1α is inhibited under hypoxia, the accumulated HIF-1α enters the nucleus, binds to another partner, HIF-1β, to form a dimer, and binds to the hypoxia response element (HRE) with the help of a coactivator cAMP response element-binding, protein-binding protein (CBP/P300) to activate the transcription and expression of target genes such as vascular endothelial cell growth factor (VEGF) and Glucose transporter 1 (GLUT1) [17].Thus, it exerts key biological functions such as promoting hypoxia-induced angiogenesis, cell metabolism, proliferation, invasion, metastasis, etc. (Figure A2) [18]. HIF-1α overexpression in tumor tissues is frequently associated with poor prognosis, implying that HIF-1α may have utility as a biomarker for assessing tumor treatment efficacy and prognosis, as well as a possible therapeutic target [19,20].

## 3. The HIF-1α Induced EMT in Hypoxia

In tumor progression, the spread and metastasis of cancer are often associated with a poor prognosis. In the study, it was found that the occurrence of EMT in many cancer cell lines is an obvious feature, which is closely related to the metastasis and spread of tumors. EMT is a cellular biological program that occurs naturally in a wide range of tissue types and developmental stages. Notably, under physiological conditions, epithelial cells rarely activate the EMT program, transforming into mesenchymal cells. However, in tumor tissue, the EMT program of epithelial cells is activated, causing cancer cells to lose cell–cell junctions and degrade local basement membranes through the elevated expression of various matrix-degrading enzymes, thereby supporting their migration and invasion as individual cells [21,22]. Cancer cells undergoing EMT have not only an altered morphology, but also altered molecular characteristics. EMT is characterized by a decrease in epithelial markers such as E-cadherin (a calcium-dependent glycoprotein that promotes intercellular adhesion and maintains epithelial integrity) and an increase in mesenchymal markers such as vimentin, N-cadherin, fibronectin, various matrix metalloproteases (MMPs), and β1 and β3 integrins [23]. Studies have shown that EMT is associated with a variety of tumor functions, including tumor initiation, malignant progression, tumor stemness, tumor cell migration, metastasis, intravascular infiltration, and resistance to therapy [4]. In tumor cells, many factors are involved in the process of regulating EMT. Exploring ways to inhibit EMT is an important measure to improve the effect of tumor therapy [24].

Numerous studies have demonstrated that hypoxia promotes tumor EMT, thereby facilitating tumor invasion and metastasis. Hypoxia can affect the EMT process by: regulating EMT-related signaling pathways such as TGFβ, WNT-β-catenin, Notch, and Hedgehog (SHH) pathways; regulating the expression of EMT transcription factors (EMT-TFs) such as TWIST1, SNAIL1, and ZEB1, ZEB2, SNAIL2 (SLUG); and regulating EMT-associated RNAs [25]. The occurrence of EMT is associated with the disruption of epithelial homeostasis through a network of mechanisms regulating transcriptional translation processes, non-coding RNA expression, selective splicing, and protein stability (Appendix A, Figure A3) [26]. Many studies have pointed out that HIF-1α is crucial for the emergence of EMT, and recent studies have shown that HIF-1α mainly promotes EMT by interacting with EMT regulators such as Snail, Slug, TWIST, and β-catenin and by enhancing the NF-κB and Wnt/β-catenin signaling pathways in hypoxia (Figure A3) [27].

Snail and Slug are a family of zinc-finger transcription factors which increase the invasiveness of tumor cells mainly by repressing the expression of the cellular tight junction protein E-cadherin and other epithelial genes such as Claudins, Occludins, and Muc1, while increasing the expression of key mesenchymal expression genes such as fibronectin (FN1) and MMP9 [28]. HIF-1α regulates Snail expression by activating the promoter through binding to two hypoxia response elements (HREs) in the snail promoter region [29,30]. In addition, HIF-1α can activate Smad3 through the TGF signaling pathway, and Smad3 regulates EMT progression by directly binding to the promoters of Snail and Slug, activating their transcription and promoting the expression of Snail and Slug [31,32]. HIF-1α promotes the expression of SNAIL and Slug, thereby accelerating the progression of EMT under hypoxic conditions and enhancing the initiation of metastasis in prostate cancer [33].

β-catenin is an important mediator in the Wnt/β-catenin signaling pathway. In normal somatic cells, β-catenin only acts as a cytoskeletal protein in the cell membrane, forming a complex with E-cadherin to maintain homologous cell adhesion and prevent cell migration. Upon entering the cytoplasm, β-catenin can be phosphorylated by GSK-3β, preventing its accumulation in the cytoplasm [34]. When cells undergo EMT, β-catenin dissociates from the complex on the cell membrane, aggregates in the cytoplasm, and translocates to the nucleus where it promotes the transcription of genes that induce EMT. HIF-1α could regulate the EMT process through the β-catenin pathway. When HIF-1α expression was repressed, the levels of Snail and β-catenin were decreased and the EMT process was inhibited [29]. In addition, cells under a hypoxic environment can exert a range of biological effects through the activation of the Wnt/β-catenin pathway. HIF-1α increases Wnt/β-catenin signaling in hypoxia embryonic stem (ES) cells by enhancing β-catenin activation and elevating downstream effector lymphoid enhancer-binding factor-1 (LEF-1) and T-cell factor-1 (TCF-1) expression [35].

TWIST1 and TWIST2 belong to the basic helix–loop–helix (bHLH) family of EMT-TFs, which regulate the expression of target genes by binding to E-box response elements. TWIST proteins are elevated during cancer progression and TWIST1 has been shown to diminish intercellular adhesion by decreasing epithelial E-cadherin expression and promoting mesenchymal Fibronectin, N-cadherin, and Vimentin expression [36]. HIF-1α directly binds to the hypoxia response element (HRE) of the TWIST proximal promoter and up-regulates TWIST 1 expression, ultimately promoting the development of EMT [34,37]. A retrospective analysis of 87 patients with non-small cell lung cancer (NSCLC) showed that overall survival was significantly shorter in patients with overexpression of HIF-1α, TWIST1, or Snail, and that co-expression of any two factors could be used as a predictive diagnostic indicator for patients [38].

ZEB proteins are members of the zinc finger E-box binding homeobox family, which is one of the most essential nuclear transcription factors. ZEB1 and ZEB2 are important transcription factors in the EMT process. As with TWIST1 and SNAIL1, the ZEB1 promoter region has regions that directly recognize and bind to HIF-1α, which binds to the hypoxia response element 3 of the ZEB1 proximal promoter and regulates ZEB1 expression [39]. The study demonstrated that when exposed to persistent hypoxia, HIF-1α up-regulates the expression of ZEB2, which promotes irreversible EMT [40]. ZEB2 acts as an upstream regulator of ETS proto-oncogene 1 (ETS1), which can initiate and maintain the EMT phenotype by mediating the expression of TWIST and MMP9 via the ZEB2/ETS1 axis [41].

Besides EMT-TFs, there are some important signaling pathways involved in the EMT process. NF-κB has been found to be a key pathway in the EMT process in mouse models, and blocking the NF-κB signaling pathway can reverse EMT to MET and significantly reduce tumor metastasis [42]. Additionally, inhibition of NF-κB decreased the expression of EMT-TFs (SNAI1, SNAI2, and ZEB1) as well as the mesenchymal markers VIM and CDH2, ultimately reducing the metastasis process in vivo [43]. According to further studies, NF-κB and HIF-1α can cooperate to activate one another. The stabilization of HIF-1α increases the activation of the NF-κB signaling pathway, which in turn enhances the production of HIF-1α, thereby promoting the development of EMT [12,44].

Additionally, HIF-1 can facilitate the occurrence of EMT by interacting with other EMT-related factors. For example, under hypoxic conditions, the expression of lysyl oxidase (LOX) and LOX-like 2 (LOLX2) genes can be enhanced by HIF-1α, and then the expression of E-cadherin can be down-regulated, resulting in the promotion of EMT and cell invasion and migration ability [45,46]. P2X receptors are a family of cation-permeable ligand-gated ion channels that open in response to extracellular ATP binding. P2X receptors are classified into seven subtypes, termed P2X1–7. P2X7 receptors are expressed prevalently in a range of tissues, including tumor tissues [47,48]. MMP-14, MMP-2, and MMP-9 are the major types of matrix metalloproteins (MMPS), which are involved in the protein degradation of the basement membrane and extracellular matrix (ECM) and play key roles in angiogenesis and metastasis. MMPs are induced by multiple growth factors and signaling pathways, including NF-κB, EGF, PI3K, and RTK [49,50]. The expression of the P2X7 receptor can be induced by elevated HIF-1α under hypoxia, and then P2X7 can phosphorylate ERK and AKT signaling pathways and increase the accumulation of NF-κB, which promotes tumor cell EMT by regulating the expression of MMP2 and MMP9 [51,52].

## 4. HIF-1α Can Promote Tumor EMT Progression by Regulating LncRNA Expression under Hypoxia

Over 75% of the human genome is transcribed into RNA, but less than 2% of the transcripts are translated into proteins. These RNAs that cannot be translated into proteins are called non-coding RNAs (ncRNAs). Long non-coding RNAs (lncRNAs) are RNAs that contain more than 200 nucleotides. The role of lncRNA has been increasingly revealed over the last few years through investigation. By combining with DNA, RNA, and protein, it can contribute the biological functions of cell metabolism, proliferation, migration and invasion, and angiogenesis. Additionally, it can be used clinically as a biomarker in the diagnosis, treatment, and prognosis of cancer [53,54].

There have been many studies on the mechanisms involved in the regulation of hypoxia by lncRNAs. Under a hypoxic microenvironment, the expression of lncRNA can be induced directly or indirectly via HIF-dependent and HIF-independent. The direct way is through binding to the target on the lncRNA promoter region, while the indirect way is through epigenetic regulation to affect RNA stability. Among them, the HIF-dependent way is more common [55]. Notably, lncRNA can be induced to be expressed in a HIF-1α-dependent manner, stimulating the production of EMT transcription factors that regulate EMT development and promote tumor metastasis, invasion, malignant proliferation, and poor prognosis [56]. Under hypoxia, HIF-1α induced by tumor cells can bind to HRE on lncRNA and promote the expression of HIF-1α antisense lncRNA (HIFAL) [57]. HIFAL participates in a variety of biological processes, such as chromatin organization, nuclear structure, and gene transcription; most HIFALs can also act as competing endogenous RNA (ceRNA) or RNA sponges to compete with miRNAs for post-transcriptional regulation, interfering with the binding of miRNA and target genes, thereby affecting the transcription of target genes and reducing the silencing effect of these miRNAs on target mRNAs [58,59]. In addition, HIFAL can accelerate cancer progression by regulating protein activity and participating in the synthesis, accumulation, and stabilization of protein complexes [53,57]. Moreover, studies have shown that prolyl hydroxylase 3 (PHD3) can be recruited by HIFAL to bind to pyruvate kinase 2 (PKM2), inducing its proline phosphorylation to form the PKM2/PHD3 complex. The PKM2/PHD3 complex is induced into the nucleus after HIFAL binds to heterogeneous nuclear ribonucleoprotein F (hnRNPF) and the transcriptional activity of HIF-1α is enhanced, forming a positive feedback pathway activated by HIF-1α [60].

As previously described, HIF-1α can promote EMT in tumor cells by directly regulating EMT transcription factors and signaling pathways, creating conditions for tumor metastasis. Similarly, some lncRNAs can be directly induced by HIF-1α and then promote tumor EMT progression. Since lncRNAs play an important role in tumors, drugs targeting lncRNAs are being developed and carried out according to the different mechanisms of action of targeting lncRNAs and different drug delivery systems (which can be mainly divided into viral vector transport and non-viral vector transport) [8]. In comparison to conventional drug therapy and protein-level therapy, however, gene therapy research targeting lncRNA is a more novel attempt that faces substantial obstacles: for instance, the lack of successful reference cases and clinical data on the safety and efficacy of lncRNA-targeted drugs. In the following, we summarize the lncRNAs that are directly regulated by HIF-1α to promote tumor EMT and their mechanisms of action to broaden the understanding of clinical tumor therapy targeting lncRNAs (Appendix B, Table A1).

### 4.1. LncRNA HOTAIR

Zhou et al. found that hypoxia-induced HIF-1α in non-small cell lung cancer specifically binds to the HRES of the *HOTAIR* promoter to promote the transcriptional activity and expression level of *HOTAIR*, and then the proliferation, invasion, and migration abilities of non-small cell lung cancer are significantly increased. *HOTAIR* can bind to miR-29b and inhibit its expression in cervical cancer cells. By targeting SP1, miR-29b can modulate PTEN expression and induce EMT via the miR-29b/PTEN/PI3K axis, and so promotes cancer cell motility, invasion, and chemoresistance to medicines such as cisplatin, paclitaxel, and docetaxel (Table A1) [61,62].

### 4.2. LncRNA RP11-390F4.3

The *lncRNA RP11-390F4.3* can be directly activated by HIF-1α under hypoxia, and the cell migration and invasion abilities are significantly reduced after silencing *RP11-390F4.3* in FADU and MCF7 cell lines. Up-regulation of *RP11-390F4.3* promotes the expression of multiple EMT regulators such as Snail, Twist1, ZEB1, and ZEB2, which promote tumorigenesis and metastasis (Table A1) [63,64].

### 4.3. LncRNA FALEC

Analysis of prostate cancer cases showed that HIF-1α can bind to the HRE of *lncRNA FALEC* in prostate cancer cells and induce the increase in *FALEC* expression. Down-regulation of *FALEC* expression may reduce tumor proliferation, migration, and invasion by reversing EMT [65]. In addition, *FALEC* expression was significantly increased in gastric cancer tissue samples, and after *FALEC* expression was silenced, the migration and invasion abilities of gastric cancer cells were inhibited due to decreased ECM1 expression in the cell line [66]. ECM1 is a marker of tumorigenesis and represents a poor prognosis of tumors. The expression of β-catenin, one of the EMT-related genes, can be regulated by ECM1 at the post-transcriptional level, thereby further promoting tumor EMT (Table A1) [67].

### 4.4. LncRNA GAPLINC

HIF-1α can transcriptionally activate *GAPLINC*, which is highly expressed in gastric cancer tissues and promotes tumor migration and invasion behavior [68]. Additionally, in hepatocellular carcinoma, increased *GAPLINC* expression is associated with distant metastases and tumor stages. *GAPLINC* knockdown dramatically decreased HCC cell proliferation, invasion, migration, and EMT processes [69]. The expression level of *GAPLINC* was investigated in colorectal cancer cell lines by Yang et al., who found that elevated *GAPLINC* was closely associated with increased lymph node metastasis. At the same time, two proteins that bind to *GAPLINC*, PTB-associated splicing factor (PSF) and non-POU-domain-containing octamer-binding (NONO) protein, were verified by RNA pull-down experiments. Subsequent experiments found that *GAPLINC* promotes the expression of SNAI2 after binding to PSF/NONO, thus promoting the EMT and invasive ability of colorectal cancer cells (Table A1) [70].

### 4.5. LncRNA HAS2-AS1

The differential expression of OSCC cells and normal tissues under normoxic and hypoxic conditions was analyzed by lncRNA microarray technology, and the results demonstrated that HIF-1α can directly bind to the *HAS2-AS1* promoter and activate the transcription of *HAS2-AS1* under hypoxia. Up-regulated *HAS2-AS1* promotes EMT and the cell invasive ability of OSCC tumor cells by stabilizing the expression of HAS2 (Table A1) [71].

### 4.6. LncRNA MALAT1

*LncRNA MALAT1* has been shown to be directly triggered and transcribed by HIF-1α upon hypoxia, resulting in elevated *MALAT1* expression. The up-regulated *MALAT1* can serve as an endogenous RNA by competitively binding to *miR-3064-5p* and promoting the proliferation and migration of breast cancer cells [72]. *MALAT1* was overexpressed in oral squamous cell carcinoma OSCC and induced the expression of EMT-related transcription factors β-catenin and NF-κB. After silencing *MALAT1*, the expression of E-cadherin was increased, while the expression of mesenchymal markers N-cadherin and Vimentin was decreased. Therefore, it is demonstrated that *MALAT1* maintains the cellular EMT phenotype and mediates the invasion and migration of cells [73]. Additionally, *lncRNA MALAT-1* inhibits the expression of the epithelial marker E-cadherin in bladder cancer cells by targeting the *miR-124*/foxq1 axis and acting as a downstream mediator of TGF-β with the suppressor of zeste 12 (suz12), while increasing the expression of the mesenchymal markers N-cadherin and fibronectin, thereby promoting the bladder cancer cells’ EMT (Table A1) [74,75].

### 4.7. LncRNA UCA1

In breast cancer cell lines, the expression of *lncRNA UCA1* was found to be significantly increased under hypoxic conditions, and the proliferative and anti-apoptotic abilities of breast cancer cells were also increased. After HIF-1α was silenced, the level of *UCA1* reduced dramatically, indicating that the expression of *UCA1* was carried out in a HIF-1α-dependent manner [76]. *LncRNA*
*UCA1* can bind to *miR-143* and block its expression, inducing increased high mobility group box 1 (HMGB1) expression. Additionally, elevated *UCA1* promotes tumor EMT progression and tumor invasion through the *miR-143*/HMGB1/*UCA1* pathway (Table A1) [77].

### 4.8. LncRNA H19

In glioblastoma cell lines under hypoxia, HIF-1α can stimulate the expression of *H19* by directly binding to the *H19* promoter, or by up-regulating the expression of specific protein 1 (SP1) protein and then binding to the *H19* promoter to promote the expression of *H19* [78]. The level of *lncRNA H19* was significantly increased in liver fibrotic cells. After further study, it was found that HIF-1α could interact with the *H19* promoter region in the HRE original at 492–499 and 515–522 bp and then promote the transcription of *H19* [79]. In addition, Corrado C et al. found that when the expression of *H19* was silenced in multiple myeloma (MM) cell lines, the transcription level of HIF-1α and the expression levels of its downstream target genes VEGF and SNAIL were reduced, suggesting that *H19* is a HIF-1α activator. Therefore, we can speculate that there may be a mutual activation pathway between *lncRNA H19* and HIF-1α, which regulates tumor growth under hypoxia through mutual promotion of expression [80]. Meanwhile, *H19* functions as a competing endogenous ceRNA with miR-138 and miR-200a to target and stimulate the expression of mesenchymal marker genes such as *Vimentin*, *ZEB1*, and *ZEB2* in colorectal cancer cells, hence promoting EMT (Table A1) [81].

### 4.9. LncRNA BACE1-AS

The HIV-1 transcriptional transactivator (Tat) protein has been shown to induce toxic neuronal amyloid production and enhance neurotoxicity. Studies have shown that HIV-1 Tat can up-regulate the expression level of HIF-1α in human primary astrocyte (HPA) and increase the translocation of HIF-1α into the nucleus, where it binds to the *lncRNA BACE1-AS* promoter, resulting in increased *lncRNA BACE1-AS* synthesis. Further analysis by means of EMSA and RIP assays proved that the HIF-1α protein can form a physical bond with *lncRNA BACE1-AS*. Through ChIP analysis, it was found that the HIF-1α and *BACE1-AS* complex can regulate the synthesis of the β-site cleaving enzyme (BACE1) at the transcriptional and post-transcriptional levels, and subsequently lead to increased BACE1 protein synthesis, which, in turn, promotes the production of amyloid β protein [82]. Additionally, *BACE1-AS* is up-regulated in the hepatocellular carcinoma cell line (HCC), and increased *BACE1-AS* can regulate the expression of the gene *CELF1* by down-regulating the level of *miR-377-3p*, resulting in the activation of the EMT pathway and increased tumor invasion and metastasis (Table A1) [83].

### 4.10. LncRNA BX111

*LncRNA-BX111* was overexpressed in pancreatic cancer tissues. After the silencing and overexpression of *BX111* experiments, it was found that the overexpression of *BX111* is beneficial to improving the proliferation and invasion ability of tumor cells. Further experiments demonstrated that under hypoxia, HIF-1α induced the transcription of *lncRNA BX111* to up-regulate its expression, and that *lncRNA-BX111* could then activate ZEB1 and its downstream proteins E-cadherin and MMP2, promote the development of EMT in pancreatic cancer cells, and improve cell metastasis and invasion (Table A1) [84].

### 4.11. LncRNA CASC9

*LncRNA CASC9* was found to be overexpressed in lung cancer tissues, and overexpressed *CASC9* significantly increased the EMT, invasion, and migration abilities of tumor cells. It was demonstrated that *CASC9* can be induced by HIF-1α and promote the stability of HIF-1α, indicating that HIF-1α and *CASC9* can form a mutual activation pathway to promote the progression of lung cancer (Table A1) [85].

## 5. Targeted Drugs That Inhibit the Biological Function of HIF-1α

Hypoxia is a hallmark of many types of solid tumors and the effects of hypoxia on tumors are mainly mediated through HIF-1α. In the above, we illustrated that HIF-1α can promote EMT progression in tumor cells by regulating the expression of EMT-TFs and HIFAL, which in turn promotes tumor invasion, migration, and distant metastatic ability. The use of HIF-1α inhibitors can attenuate tumor EMT, migration, invasion, and metastasis by blocking the regulatory effects of HIF-1α on EMT-TFs and HIFAL. Given that HIF-1α plays a critical role in tumor formation, research into targeted HIF-1α therapy is presently a trending topic. Currently, drug research is mainly carried out by interfering with HIF-1α transcription, HIF-1α mRNA translation, HIF-1α protein degradation, and HIF-1α targeting genes. Through research, drugs have been shown to have good anti-HIF-1α effects and are practiced in clinical treatment. These drugs can be divided into natural drugs and artificial chemical synthesis drugs. Next, we will classify and summarize the current HIF-1α-targeting inhibitor drugs according to their mechanism of action from the following aspects, aiming to provide clinical treatment ideas for reversing poor tumor progression by inhibiting HIF-1α production and accumulation.

### 5.1. HIF-1α Inhibitors That Inhibit HIF-1α mRNA Levels

KUSC-5001 is a novel HIF-1α inhibitor discovered by Marina Sakai et al. The effective molecule contained in KUSC-5001 is 1-alkyl-1H-pyrazole-3-carboxamide, which inhibits the level of HIF-1α mRNA by acting on the target protein ATP5B (the catalytic β subunit of mitochondrial FoF1-ATP synthase) and reduces HIF-1α target genes such as carbonic anhydrase 9 (CA9) and VEGF gene levels, thereby inhibiting the transmission of HIF-1α downstream signaling pathways. There are limited studies on the use of KUSC-5001 as a HIF-1α inhibitor, so it needs to be further explored as a possible anti-cancer agent [86].

Flavopiridol is a novel cyclin-dependent kinase inhibitor, which has been found to interfere with HIF-1α gene transcription and down-regulate HIF-1α mRNA expression in the presence of proteasome inhibitors. It can further inhibit tumor vascularization by reducing the expression of the HIF-1α target gene VEGF [87].

EZN-2968 is an RNA antagonist consisting of a third-generation oligonucleotide, lock nucleic acid, that specifically binds and inhibits the expression of HIF-1α mRNA. EZN-2968′s potent, selective, and sustained inhibition of HIF-1α mRNA and protein levels and tumor cell growth has been validated in prostate cancer and glioblastoma cell lines [88].

S-2-amino-3-[4-N,N,-bis(2-chloroethyl)amino] phenyl propionic acid N-oxide dihydrochloride (PX-478) has been shown to inhibit the expression levels of HIF-1α and HIF-1α downstream target genes VEGF and GLUTT-1. Although the specific mechanism has not yet been elucidated, it has been found that the current mechanisms by which PX-478 reduces HIF-1α levels include: reducing HIF-1α mRNA levels, inhibiting HIF-1α translation, and inhibiting HIF-1α de-ubiquitination to increase protein degradation. Among them, the reduction of the HIF-1α mRNA level and translation has been shown to play a major role, while the inhibition of HIF-1α de-ubiquitination plays a minor role [89,90,91]. The efficacy of PX-478 against non-small cell lung cancer was determined by studying several lung cancer cell lines and orthotopic mouse models [92]. HIF-1α was found to be overexpressed in esophageal squamous cell carcinoma (ESCC), and in vitro and in vivo experiments demonstrated that PX-478 has significant antitumor activity against HIF-1α-overexpressing ESCC tumors [93].

### 5.2. HIF-1α Inhibitors That Inhibit HIF-1α Transcriptional Activity

Acrifavine (ACF) has trypanocidal, antibacterial, and disinfectant properties and can be used to treat gonorrhea. Parallelly, as an FDA-approved small-molecule HIF-1α inhibitor, acrifavine inhibits tumor development by disrupting HIF-1α and HIF-1β dimer formation, inhibiting the transcriptional activity of HIF-1α, and down-regulating the expression of its downstream proteins VEGF and PGK-1, thereby inhibiting the signaling of downstream pathways. Studies have shown that acrifavine can successfully inhibit brain cancer tumor growth [94,95].

Aminoflavone (AF) is the active component of a novel anticancer agent (AFP464) in phase I clinical trials and is a ligand for the aryl hydrocarbon receptor (AhR). There have been many studies on the anticancer effects of AF, such as by Lancelot McLean et al., who showed that AF can inhibit the growth of breast cancer cells in part by inducing ROS production, oxidative DNA damage and apoptosis, and has the potential to treat hormone-dependent breast cancer [96]. It has been demonstrated experimentally that aminoflavone inhibits HIF-1α transcriptional activity and protein accumulation in breast cancer cell lines, and the inhibitory effect of AF on HIF-1α is independent of AhR function. However, it is worth noting that this effect has cell line limitations—for example, AF inhibited HIF-1α accumulation in the MDA/SULT1A1 cell line but had no significant effect in the MDA-MB-231 cell line. Therefore, aminoflavone should be considered, when used as a HIF-1α inhibitor, as being limited by cell type [97].

Echinomycin (EKN), an HIF-1α inhibitor, inhibits HIF-1α and DNA binding to reduce HIF-1α transcriptional activity in the presence of hypoxia. A significant reduction in hypoxia-dependent cellular recovery was found in wound-healing assays in EKN-treated adult retinal pigment epithelium (aRPE) cells. Reduced levels of HIF-1α mediated transcripts were detected in hypoxic aRPE cells compared to untreated control cells, and HIF-1α-dependent angiogenesis was successfully inhibited in an in vitro mouse model [98]. Studies have further indicated that the in vivo efficacy of echinomycin in HIF-1α overexpressing solid tumors was formulation dependent. Compared to the previously used Cremophor formulation of echinomycin, liposome-echinomycin showed significantly enhanced inhibition of primary tumor growth and reduced established triple-negative breast cancer (TNBC) metastases [99]. Benoit Vlaminck’s team noted that echinomycin increased HIF-1α activity under normoxia and that it had a dual effect on HIF-1α activity, thus suggesting that the specificity of echinomycin as a cancer therapeutic agent is limited [100].

By analyzing the clinical drug library, the experimental team found that anthracycline chemotherapeutic drugs doxorubicin and daunorubicin may be HIF-1α inhibitors. They can reduce HIF-1α transcription by blocking its binding to DNA and inhibit the proliferation and angiogenesis of prostate cancer by inhibiting HIF-1α target gene expression [101].

Bortezomib, the first proteasome inhibitor, is clinically available for the treatment of hematologic tumors. It has been shown that bortezomib inhibits HIF-1α transcriptional activity and attenuates its protein synthesis and the expression of its nuclear targets such as VEGF by inhibiting the PI3K/Akt/mTOR and MAPK pathways in AD and AI prostate cancer (PCa) cells, respectively [102].

### 5.3. HIF-1α Inhibitors That Inhibit HIF-1α Protein Synthesis and Accumulation

It was found that 7-Hydroxyneolamellarin A (7-OH-Neo A), a natural product derived from the ocean, may target proteins related to HIF-1α stabilization under hypoxia, inhibiting HIF-1α protein accumulation and the transcription of its downstream target gene VEGF activity and showing a good inhibitory effect on the HIF-1α pathway. It is worth noting that the inhibitory effect of 7-OH-Neo A on HIF-1α may not depend on the reduction of HIF-1α transcription, translation, and protein degradation. Experiments have shown that the7-OH-Neo A has anti-tumor activities such as anti-tumor angiogenesis, proliferation, invasion, and migration, and can be used as a promising HIF-1α inhibitor in the future [103].

Melatonin (N-acetyl-5-methoxytryptamine) is an endogenous hormone produced by the pineal gland and has been shown in many studies to reduce the expression of HIF-1α [104]. Studies have demonstrated that melatonin decreased the level of HIF-1α protein but not mRNA in the cells and additionally decreased the level of HIF-1α target gene expression after the administration of melatonin in kidney HK-2 cells [105]. In addition, another team has demonstrated that N-butyryl-5-methoxytryptamine (NB-5-MT), a derivative of melatonin, can reduce the expression of HIF-1α protein and the transcription of HIF-1α target genes, and has a high strong anti-angiogenesis effect in tumors, which can be used as a potential new anti-tumor drug [106].

A number of studies have reported that H2S can regulate the expression of HIF-1α and may exert beneficial effects by inhibiting HIF-1α production in some diseases [107]. The team used two H2S donors, namely sodium hydrosulfide (NaHS) and sodium sulfide (Na2S), and demonstrated experimentally that H2S donors under hypoxia can inhibit HIF-1α protein expression and HIF-1α-dependent gene expression, and that this inhibition is VHL-dependent [108]. GYY4137, a slow-releasing H2S donor, has been shown to inhibit the proliferation and angiogenesis of cancer cells in HCC cell lines by directly inhibiting the activation of the STAT3 pathway, which in turn inhibits its downstream target proteins, including HIF-1α, and effectively suppresses the expression levels of HIF-1α and its downstream gene VEGF [109].

There have been many studies on the anticancer effects of ATR inhibitors. In tumor hypoxia, the cellular stress response causes the activation of ATR and induces the translation of HIF-1α mRNA to promote cellular adaptation to hypoxia, and thus, the application of the ATR inhibitor can reduce the expression of HIF-1α protein and inhibit the expression level of its downstream target genes such as GLUT-1. For example, AZD6738 can control the proliferation of HCC cells by improving the immune microenvironment, prolong survival, and prevent tumor recurrence in HCC patients. Another promising anticancer drug, VX-970, can be used as a monotherapy or in combination with other drugs for chemotherapy and radiotherapy [110,111,112]. In addition, experiments demonstrated that VE-821 can inhibit ATR-mediated signaling under hypoxia, thereby inhibiting the stability of HIF-1α and increasing the sensitivity of tumor cells to radiotherapy [113].

Histone Deacetylase Inhibitors (HDACi) are a class of inhibitors that interfere with the function of histone deacetylases. Studies have shown that HDACi can inhibit tumor angiogenesis by inhibiting the expression of HIF-1α. The study further proved that HDACi may inhibit the translation process of HIF-1α protein by inhibiting the signaling molecules PI3K and GSK3β, thereby inhibiting the expression of HIF-1α protein [114]. Suberoylanilide Hydroxamic Acid (SAHA), an HDACi that has been approved for the treatment of cutaneous T cell lymphoma (CTCL), has been found to down-regulate HIF-1α protein levels. Interestingly, SAHA does not reduce HIF-1α mRNA levels, but inhibited HIF-1α protein expression levels by silencing HDAC9 to prevent post-translational acetylation of HIF-1α and interaction with the Hsp70/Hsp90 chaperone axis. In the future, the molecular mechanism of HDACi’s inhibition of HIF-1α expression needs to be further explored [115].

Camptothecin, an alkaloid extracted from the Hippophae tree, is a natural inhibitor of DNA topoisomerase-I. CRLX101 is a nanoparticle-drug conjugate containing CPT, which clinical trials have shown to be a promising radiosensitizer for improving CRT in rectal cancer. CRLX101 has a promising toxicity profile, inhibits HIF-1α up-regulation in a prolonged manner, and leads to reduced expression of downstream HIF-1α signaling targets such as CAIX and VEGF by inhibiting HIF-1α. However, the exact molecular mechanism has not been elucidated yet and further studies are needed [116,117,118].

When studying the effect of topoisomerase I inhibitor topotecan (TPT) on VEGF induced by hypoxia in human neuroblastoma cells, it was found that TPT, a camptothecin analogue, can inhibit the synthesis and transactivation of HIF-1α protein, and thereby inhibits the expression of HIF-1α target gene VEGF, shows certain anti-angiogenic activity, and inhibits the angiogenesis and growth of neuroblastoma [119]. By establishing a rabbit model of pulmonary hypertension, it was found that TPT could improve hypoxia-induced pulmonary vascular remodeling in PAH by inhibiting the expression of HIF-1α [120].

EZN-2208 is a camptothecin analogue used in clinical trials for solid tumors and lymphomas. It was shown that EZN-2208 can effectively inhibit the expression of HIF-1α protein and the mRNA expression levels of HIF-1α target genes such as MMP2, VEGF1, Glut1, Glut3, and TGFβ1 in mice. As a result, tumor metabolism, angiogenesis, and cell migration are diminished [121,122]. In addition, experiments have confirmed that EZN-2208 can show significant anti-early leukemic activity in chronic lymphocytic leukemia (CLL) by acting as an HIF-1α inhibitor [123].

Ellagic acid (EA), a naturally occurring polyphenolic compound found in many vegetables and fruits, reduces the protein expression and activation of HIF-1α and the transcriptional levels of HIF-1α-targeted genes in lung cancer cells [124]. By establishing a rat arthritis model, it was found that EA significantly reduced the levels of HIF-1α and VEGF proteins in the synovial membrane of adjuvant arthritis rats, showing good anti-synovial angiogenic effects [125].

Similarly, the natural product Bufalin is a traditional analgesic drug, which has been found to have anticancer activity in various tumors in recent years. For example, bufalin can down-regulate the protein expression level of HIF-1α in ovarian cell carcinoma. In studies of hepatocellular carcinoma, bufalin was found to down-regulate HIF-1α expression by inhibiting the PI3K/AKT/mTOR signaling pathway, and also to reduce cell invasion by inhibiting the TGF-β-induced EMT process [126,127].

### 5.4. HIF-1α Inhibitors That Promote Proteasomal Degradation

The chaperone heat shock protein 90 (HSP90) has been shown to stabilize the HIF-1α subunit, protecting it from proteasomal degradation. At present, HIF-1α can be destabilized by drugs that inhibit HSP90. Studies have shown that HSP90 inhibitor 17-allylamino-17-demethoxygeldanamycin (17-AAG) can inhibit the production of HIF-1α protein and target genes in multiple myeloma under hypoxia, indicating that it may serve as an HIF-1α inhibitor against cancers. Additionally, it was demonstrated that the HSP90 inhibitor ganetespib inhibits the interaction between HSP90 and HIF-1α, hence decreasing HIF-1α and VEGF expression [128,129]. In addition, another HSP90 inhibitor, AT-533, can reduce the mRNA and protein levels of HIF-1α and inhibit the expression of its downstream target gene VEGF and downstream signaling pathways [130]. Glyceollins, plant antitoxins isolated from soybean, were found to reduce HIF-1α levels in a variety of tumor cell lines by inhibiting the PI3K/AKT/mTOR pathway to block HIF-1α translation and by inhibiting the binding of HIF-1α to Hsp90 to promote HIF-1α degradation, thereby reducing the stability of HIF-1α [131].

Cetuximab, a monoclonal antibody to EGFR, has been shown to inhibit the accumulation of HIF-1α protein by inhibiting the activation of EGFR downstream signaling pathways (including Erk, Akt, and mTOR) and promoting HIF-1α degradation that depends on the prolyl hydroxylase (PHD) pathway under hypoxia [132]. Another study found that cetuximab inhibited glycolysis in cetuximab-sensitive head and neck squamous cell carcinoma (HNSCC) cells by down-regulating the expression of HIF-1α and suppressing the expression of the downstream target gene lactate dehydrogenase A (LDH-A) [133].

LW6 was identified as a novel small compound that inhibits the accumulation of HIF-1α, and it was shown that LW6 induces the expression of VHL and promotes its protein degradation by binding to HIF-1α. Therefore, LW6 can reduce the expression of HIF-1α protein by affecting its stability, and is a promising HIF-1α inhibitor to be developed [90,134].

Metformin (Met), a conventional drug for the treatment of diabetes, has been shown in numerous studies to be an effective HIF-1α inhibitor in cancer therapy, where it can reduce HIF-1α expression by up-regulating the p-AMPK pathway in cancer cells and inducing the expression of PHDS, which results in HIF-1α degradation by the proteasome. In addition, by exploring the protective effect of Met on murine liver injury in a murine model, the results showed that metformin might protect rats from thioacetamide-induced liver injury by inhibiting the mTOR–HIF-1α axis [135]. By evaluating the effect of Met on Angiopoietin-like 4 (ANGPTL4) induced by nickel exposure and the role of ANGPTL4 in lung carcinogenesis, it was found that HIF-1α accumulation induced by nickel in lung cells leads to an increased presence of ANGPTL4, while the use of Met attenuated the NiCl_2_-induced ANGPTL4 production by inhibiting HIF-1α expression and its binding activity [136]. In a mechanism of action study of metformin in hypoxia-induced EMT in keloid fibroblasts (KFs), it was shown that metformin also inhibited hypoxia-induced EMT in KFs through inhibition of the HIF-1α/PKM2 signaling pathway [137]. These results illustrate the promise of Met as an HIF-1α inhibitor in oncology treatment [138].

Ginsenoside 20(S)-Rg3 is a pharmacologically active ingredient extracted from ginseng, which has been studied to reduce the expression level of HIF-1α by promoting the degradation of HIF-1α proteasome through the ubiquitin-proteasome pathway. It has also been shown to reduce the level of EMT transcription factor Snail in ovarian cancer cells, thereby inhibiting the development of EMT in tumor cells and thus reducing the metastatic ability of tumors [139].

Furthermore, as described above, HDACi is able to mediate HIF-1α proteasomal degradation independent of the pVHL pathway by promoting heat shock protein-70 (HSP70) expression, which ultimately impairs tumor growth [114].

### 5.5. HIF-1α Inhibitor That Inhibits the Formation of HIF-1-a/p300 Complex

Chetomin is a metabolite complex produced by several fungi of the genus Chaetomium that can target and block the formation of the HIF-1α/p300 complex and inhibit the transcriptional activity of genes involved in the proliferation, survival, and development of tumor cells in response to hypoxia. It has been shown to enhance anticancer activity in a variety of myeloma cell lines [140]. Several members of the epithiodipiperazine (ETP) family of natural products that can disrupt the HIF-1α/p300 complex in vitro were explored, including gliotoxin, chaetocin, and Chetomin. Experiments demonstrated that these ETP members inhibited the formation of the HIF-1α/p300 complex in prostate cancer cell lines and colon cancer cell lines and attenuated HIF-1α transcription of the downstream target genes VEGF, lactate dehydrogenase A (LDHA), and enolase-1 (ENO1), thereby effectively inhibiting tumor angiogenesis and growth [141].

The aminocoumarin antibiotic, novobiocin, is an HSP90 c-terminal inhibitor that directly blocks the protein–protein interaction between the HIF-1α C-terminal activation domain (CTAD) and the cysteine-histidine-rich histidine (CH1) region of p300/CBP. In addition, novobiocin can down-regulate the expression of genes regulated by HIF-1α, especially CA9, which is related to tumorigenesis, and thus exert an inhibitory effect on tumor cell proliferation [142,143].

## 6. Conclusions

Hypoxia is a common and important feature of many solid tumors due to rapidly growing tumor tissue with uneven angiogenesis. Tumors in hypoxic environments activate the expression of a series of target genes through the induction of HIF-1α production, which ultimately affects tumor metabolism and angiogenesis and promotes increased proliferation, metastasis, migration, and invasion of tumor cells. EMT is a common state of tumor cells, and reversing the EMT state of tumors is a hot topic in tumor therapy. HIF-1α causes the up- or down-regulation of various lncRNA levels in cells, and the altered lncRNA levels are involved in regulating EMT and promoting the malignant progression of tumors. Therefore, this review illustrates the mechanism of action of HIF-1α production and regulation of EMT occurrence by regulating lncRNA levels, and we will continue to enrich and update the knowledge on this topic in the future as the research progresses. HIF-1α, as a core substance, can regulate tumor progression in several ways, and so the study of HIF-1α inhibitors becomes more and more important; we provide a brief summary for the HIF-1α inhibitors developed so far. The design of HIF-1α inhibitors is not easy because of the numerous molecular pathways involved. At present, most of the inhibitors have been studied only at the cellular level or in animal models, and those that are actually used in the clinic are still very limited. We believe that as the research continues, the future prospects of HIF-1α inhibitors in clinical oncology applications are very optimistic.

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
