# Peer review of "Regulating the Expression of HIF-1α or lncRNA: Potential Directions for Cancer Therapy"

_cells, 2022, doi:10.3390/cells11182811_

Round 1

Reviewer 1 Report

The authors present an overview of two important topics in the field of hypoxia: promotion of EMT, particularly through lncRNAs, and potential drugs that target HIF-1 signaling. However, there is currently not a common thread between these two ideas. If the authors do not want to split these into two reviews, they need to better highlight how these two ideas are connected.

Major Comments

1.     Section 4 (drugs targeting EMT) has nothing to do with hypoxia or lncRNAs and should be deleted.

2. In Section 5, the authors list several genes as lncRNAs that are not lncRNAs (HOXA9 and EFNA3). The authors need to use the correct terminology for genes. Additionally, one of the papers they cite in this section has been redacted (reference 64). The authors either need to either cite another paper with these findings or remove all text regarding these findings from the manuscript.

3. The last paragraph of subsection 5.14 has nothing to do with this particular lncRNA. This either needs to be its own subsection or needs to be moved to the introduction part of section 5.

4. Subsection 6.1 purports to be about drugs that block HIF1A transcription, but it describes several drugs that actually block HIF-1 activity (i.e., transcriptional activation by HIF-1). These should be in their own subsection or described properly as a different class of drug.

5.  The authors use HIF-1 and HIF-1α interchangeably. These are not interchangeable, and this may confuse readers unfamiliar with HIF-1 signaling. This needs to be fixed.

6. The authors should describe how HIF-1α binds to the lncRNA BACE1-AS to promote the RNA-RNA complex (lines 351-354), as most readers are likely only aware of HIF-1 as a transcription factor.

7. The authors state the effects of PX-478 on HIF-1α but not the underlying cause of these effects. If the PX-478 mechanism/target is known, this should be described. (If unknown, this should be stated.)

8. The authors should provide a brief background on EMT and its import in cancer.

9. The language in the text needs to be significantly revised

Minor revisions

1.     What is meant by HIF-1α being a progenitor (line 30)?

2.     Lines 49/51 should state HIF-2α (not HIF-2a)

3.     Line 154 should read LOXL2

4.     The drug discussed in lines 461/462 is bortezomib

5.     Abbreviations need to be properly expanded throughout

6.     Line 591 should be degradation by the proteasome, not proteases

7.     There are multiple instances in the text where first name are included in a citation, e.g., line 105 “Astrid Brretzen et al showed …”

8.     Italics to delineate DNA/RNA need to be included.

9.     Lines 612 and 614 should read HIF-1α (not HIF-1-a)

Reviewer 2 Report

In this review, the authors well summarized how HIF-1α is induced under hypoxia and how it contributes to the development of cancer by promoting EMT. They also discussed the links between HIF-1α and lncRNAs in the process of EMT progression, as well as target drugs that are used to inhibit the EMT process and the biological function of HIF-1α. Overall the manuscript is well written and provides rich information to the authors interested in the field. I just have a few minor comments as follows:

1. The figures should be self-explanatory without the main text. Therefore it's better to provide the full names of the acronyms/abbreviations shown in the figures in the figure legends. In the current manuscript some of these full names are missing: bHLH and PAS in Fig.A1; CBP/P300 and HRE in Fig.A2.

2. Also some of the acronyms/abbreviations are not explained in their first appearance in the manuscript: 

line 15-16, EMT and lncRNAs;

line 130, NSCLC;

line 190, HCC;

line 209, ncRNAs;

line 226, ceRNA;

line 376, HNSCC.

3. In lines 71-73 and also in the figure legend of Fig.A2, the text said "...HIF-1α binds to another partner, HIF-1β, enters the nucleus...", which seems to suggest HIF-1α binds to HIF-1β before entering the nucleus. However, in Fig.2A the schematic shows that the formation of the dimer happens in the nucleus. It's confusing to the readers. Please adjust the text and/or the figure accordingly.

4. Both sections 4 and 6 are about targeted drugs. From my understanding section 4 is mostly about drugs that inhibit EMT via mechanisms other than HIF-1α, which seems to be off-topic from the rest of the review. Also it's a bit weird to talk about drugs in section 4 and then discuss the relationship between HIF-1α and lncRNAs in section 5, and then go back to drugs in section 6 again (where no drugs targeting the aforementioned lncRNAs are discussed). So I would recommend the authors shorten section 4 and merge it with section 6, and then put them both either after section 5 (so that the molecular mechanisms are discussed together and concluded with the target drugs) or before section 5 (so that lncRNAs can be emphasized to be novel drug targets).
